# The Design of an Intelligent Lightweight Stock Trading System Using Deep Learning Models: Employing Technical Analysis Methods

**SeongJae Yu [1], Sung-Byung Yang [2] and Sang-Hyeak Yoon [3],***

[1] Department of Big Data Analytics, Graduate School, Kyung Hee University, 26 Kyungheedae-ro, Seoul 02447, Republic of Korea; yousong4243@naver.com

[2] Department of Business Administration/Big Data Analytics, School of Management, Kyung Hee University, 26 Kyungheedae-ro, Seoul 02447, Republic of Korea; sbyang@khu.ac.kr

[3] School of Industrial Management, Korea University of Technology and Education, 1600 Chungjeol-ro, Cheonan-si 31253, Republic of Korea

* Correspondence: yoonsh@koreatech.ac.kr

**Abstract:** Individual investors often struggle to predict stock prices due to the limitations imposed by the computational capacities of personal laptop Graphics Processing Units (GPUs) when running intensive deep learning models. This study proposes solving these GPU constraints by integrating deep learning models with technical analysis methods. This integration significantly reduces analysis time and equips individual investors with the ability to identify stocks that may yield potential gains or losses in an efficient manner. Thus, a comprehensive buy and sell algorithm, compatible with average laptop GPU performance, is introduced in this study. This algorithm offers a lightweight analysis method that emphasizes factors identified by technical analysis methods, thereby providing a more accessible and efficient approach for individual investors. To evaluate the efficacy of this approach, we assessed the performance of eight deep learning models: long short-term memory (LSTM), a convolutional neural network (CNN), bidirectional LSTM (BiLSTM), CNN Attention, a bidirectional gated recurrent unit (BiGRU) CNN BiLSTM Attention, BiLSTM Attention CNN, CNN BiLSTM Attention, and CNN Attention BiLSTM. These models were used to predict stock prices for Samsung Electronics and Celltrion Healthcare. The CNN Attention BiLSTM model displayed superior performance among these models, with the lowest validation mean absolute error value. In addition, an experiment was conducted using WandB Sweep to determine the optimal hyperparameters for four individual hybrid models. These optimal parameters were then implemented in each model to validate their back-testing rate of return. The CNN Attention BiLSTM hybrid model emerged as the highest-performing model, achieving an approximate rate of return of 5 percent. Overall, this study offers valuable insights into the performance of various deep learning and hybrid models in predicting stock prices. These findings can assist individual investors in selecting appropriate models that align with their investment strategies, thereby increasing their likelihood of success in the stock market.

**Keywords:** attention mechanism; stock forecasting; deep learning; technical analysis method; lightweight automated stock trading system

## 1. Introduction

Individual investors face various obstacles that can hinder their success in the stock market. These barriers include emotional decision-making, difficulty understanding market trends, lack of professional expertise, and challenges in making real-time trading decisions [1]. To overcome these obstacles, quantitative automated trading systems are often employed, offering several advantages that enhance the investment experience for individual investors. These benefits encompass eliminating emotional bias, precise trade

execution through data and algorithmic analysis, and adapting swiftly to real-time market fluctuations. Moreover, as highlighted by Arribas et al. [2], these automated systems can effectively integrate investors' preferences, particularly concerning ESG objectives for diverse stocks. By leveraging these advantages, investors can optimize their investment processes, manage risks effectively, and increase their potential for sustained success in the stock market [3–5].

Despite these advantages, automated stock trading systems provided by various companies also come with a range of limitations, including potentially misleading advertisements, risks to personal information security, steep minimum investment requirements, and skepticism regarding the legitimacy of advertised returns without back-testing validation. Additionally, users often encounter significant fees when using these systems. Furthermore, there is a lack of studies that explore automated stock trading systems beyond these limitations [6]. Therefore, more studies are needed to propose methodologies that utilize deep learning models to analyze a selected set of stocks identified through technical analysis methods, followed by the implementation of automated trading [7]. These gaps in the literature emphasize the significance and potential contribution of this study.

In most previous studies, the dominant approach for predicting stock prices has been the use of Artificial Intelligence (AI) or deep learning models alone [8]. However, individual stock investors who rely solely on deep learning models for analysis and decision-making may face challenges, especially if their personal laptop's Graphics Processing Units (GPUs) have subpar performance. This is attributed to the extensive computational processing time required by deep learning models, which heavily depend on large-scale datasets. Consequently, it may be impractical to analyze all stock prices using a personal laptop GPU. On an average computer, it would take approximately three minutes to analyze a single stock for 100 epochs. Considering that there are 2546 stocks listed on the Korea Composite Stock Price Index (KOSPI), the Korea Securities Dealers Automated Quotations (KOSDAQ), and the Korean New Exchange (KONEX) markets, it would take roughly 7638 min or approximately 127 h to predict the prices of all stocks. This renders it virtually impossible to analyze all stock items and make buying and selling decisions within a single day.

To address the performance limitations of personal laptop GPUs, it is recommended to combine deep learning models with technical analysis methods. This hybrid approach can significantly reduce the required analysis time, allowing individual investors to quickly identify stocks that may yield potential gains or losses in the upcoming trading day. Therefore, this method can effectively overcome limitations associated with personal laptop GPU performance, resulting in more reliable outcomes for investment decision-making.

In this study, we propose a comprehensive buy-and-sell algorithm designed to be compatible with the performance of an average laptop GPU. This allows individual investors to implement an AI-based automated trading system across all stocks, enhancing their investment strategies and increasing efficiency. Additionally, by integrating deep learning models with technical analysis methods, investors can navigate the complexities of financial markets more adeptly and strengthen their investment strategies, ultimately achieving a stable rate of return. As a result, this approach provides a lightweight analysis method that focuses on stocks identified through technical analysis methods, rather than a large-scale analysis method that relies solely on deep learning to evaluate all stocks. This streamlined methodology saves a significant amount of time, enabling the analysis of stock items and the execution of buying and selling decisions, even with the performance capabilities of an average laptop GPU.

Despite utilizing high-speed GPUs for stock price predictions, AI-driven automated trading systems companies are marred by disadvantages, such as misleading advertising and exorbitant fees [9]. Therefore, creating a personalized automated trading system can enhance trustworthiness and provide greater autonomy, empowering individual investors to implement their preferred investment methodologies. In response to this need, we introduce a customized automated trading system tailored to investors of all scales. This

system operates within the computational capacities of an average laptop GPU to execute buy and sell orders. Given the demand for an affordable and reliable stock prediction system capable of generating steady profits within a standard laptop GPU environment, this proposal aims to deliver such a system.

Therefore, the objective of this study is to develop a system that can accurately anticipate stock market trends and yield consistent returns without relying on expensive high-performance GPUs. By leveraging the computational capabilities of average laptop GPUs, this innovative system aspires to offer individual investors a cost-effective and accessible tool for achieving their investment goals. By combining deep learning models with technical analysis methods and optimizing the system for average laptop GPUs, the findings of this study are expected to contribute valuable insights to the field of automated stock trading systems and provide practical guidance to individual investors.

## 2. Conceptual Background

### 2.1. Quant Investing

Quantitative investing, commonly referred to as *quant investing*, is prevalent in high-frequency trading and traditional asset management [10]. In traditional asset management, quant-driven investment strategies are employed to manage portfolios of stocks, bonds, and other financial assets. Investment decisions are grounded in quantifiable factors such as historical stock performance, economic indicators, and market trends. Quantitative investment strategies can range from passive investing, where the algorithm merely tracks an index, to active investing, where the algorithm seeks to generate returns exceeding the market average [11]. Although primarily utilized by institutional investors and hedge funds, quant investing is progressively becoming more accessible to individual investors. Large institutional investors, such as hedge funds and pension funds, traditionally implement these strategies, as they possess the necessary resources to invest in the technology and expertise required to develop and employ these models. The quant investment process generally involves collecting and analyzing substantial volumes of financial and economic data, developing call models to pinpoint investment opportunities, and executing trades based on the predictions of these models [5,12].

### 2.2. Previous Studies

Recently, there has been a steady stream of research on using deep learning to predict stock prices. For example, Lu et al. [13] proposed a novel stock price forecasting approach using a hybrid model, which combines the advantages of the deep learning model based on long short-term memory (LSTM) with the feature extraction capabilities of convolutional neural networks (CNNs). Their study compared multiple models in predicting stock prices using historical daily stock data spanning. Experimental results indicated that the CNN-LSTM hybrid model achieved the highest prediction accuracy, evidenced by a minimum mean absolute error (MAE) of 27.564 and root mean absolute error (RMSE) of 39.688. These metrics underscore the superior forecasting accuracy of the CNN-LSTM hybrid model compared to the other five methods [13].

Qiu et al. [14] introduced an Attention-based LSTM model integrated with wavelet transform for stock price prediction. By employing LSTM and an Attention mechanism, the wavelet transform was used to remove noise from historical stock data, extract features, and train the model. The model's performance was benchmarked against LSTM, LSTM with wavelet denoising, and the gated recurrent unit (GRU) neural network model using S&P 500, DJIA, and HSI datasets. The attention-based LSTM model with wavelet transform surpassed the other models, achieving a coefficient of determination greater than 0.94 on the S&P 500 and DJIA datasets [14].

Zaheer et al. [15] proposed an RNN model to predict next-day closing and high prices using Shanghai Composite Index data. The model's performance was evaluated against existing methods such as CNN, LSTM, CNN-RNN hybrid, and CNN-LSTM hybrid models. According to the results, CNN exhibited the poorest performance. While LSTM

outperformed the CNN-LSTM hybrid model, the CNN-RNN hybrid model surpassed the CNN-LSTM hybrid model, and the proposed RNN model outshone all other models [15].

Dai et al. [16] presented a multi-layered bidirectional LSTM (Bi-LSTM) model integrated with an Attention mechanism for predicting stock market movements, addressing forward and backward relationships. Compared to baseline models such as LSTM, BiLSTM, the classic statistical method of linear regression, Facebook's time series forecasting tool, and ARIMA, the attention-BiLSTM model demonstrated superior performance, achieving a coefficient of determination of 0.994. Furthermore, the attention-BiLSTM model showcased a lower MAE value than other models [16].

Unlike previous studies, this research is characterized by evaluating predictive performance and verifying and comparing returns through back-testing. This approach provides a more comprehensive evaluation, allowing us to confirm the prediction model's performance in the actual market. Furthermore, back-testing simulates how well a predictive model works using historical data, enabling the comparison and analysis of the model's returns. Therefore, this back-testing approach aims to verify the adequate profitability of the prediction model and derive more reliable results by comparing them with previous studies.

### 2.3. Hybrid Model Approaches

A hybrid model for predicting stock prices is a combination of multiple prediction models that aims to improve the accuracy of stock price predictions by leveraging the strengths of different models. The idea behind this approach is that individual models may have limitations, but when combined, they can produce more accurate results. The hybrid model combines various prediction models, such as time series analysis, decision trees, and artificial neural networks, to make stock price predictions [17,18].

Aldhyani and Alzahrani [17] utilized a CNN-LSTM hybrid model to predict the closing prices of stocks, assessing their performance on Tesla and Apple stock data. Their findings indicated that the CNN-LSTM hybrid model surpassed the LSTM model regarding prediction accuracy. In addition, Shi et al. [18] proposed an integrated model utilizing attention-focused CNN, LSTM, and XGBoost techniques for stock market prediction. The Attention mechanism, a neural network component, enables the model to concentrate on the most pertinent data while disregarding irrelevant information. Within the realm of stock prediction, the Attention mechanism within the hybrid model allows selective focus on the most relevant features and patterns in stock prices. This mechanism enhances the model's prediction accuracy by ensuring the model is not swayed by irrelevant or noisy data. Initially, a CNN is employed to extract relevant features from the stock prices, followed by LSTM networks that capture the temporal dependencies among the features. The Attention mechanism is then applied to these features, emphasizing the most crucial information for stock prediction [18].

The CNN-LSTM hybrid model, while effective in time series prediction and sequence data processing, has some limitations. These include high computational complexity, susceptibility to the vanishing gradient problem, the need for extensive and diverse datasets, a tendency for overfitting, lack of interpretability, and long training times. Therefore, to address these issues in this study, the author has opted for the BiLSTM layer instead of the traditional LSTM. This decision aims to improve computational complexity and enhance the model's overall performance [17,18].

### 2.4. Comparative Analysis of Hybrid Models

In recent years, deep learning models have achieved considerable prominence in a range of applications, especially within the realms of natural language processing (NLP) and prediction. Among these, hybrid models that amalgamate the strengths of various architectures—namely the convolutional neural network (CNN), bidirectional long short-term memory (BiLSTM) network, and the Attention mechanism—are particularly noteworthy. As visualized in Figure 1 and detailed in references [19,20], the distinctions among

the three models, BiLSTM Attention CNN, CNN BiLSTM Attention, and CNN Attention BiLSTM, primarily hinge on the sequencing of the layers and the specific junctures where the Attention mechanism is applied.

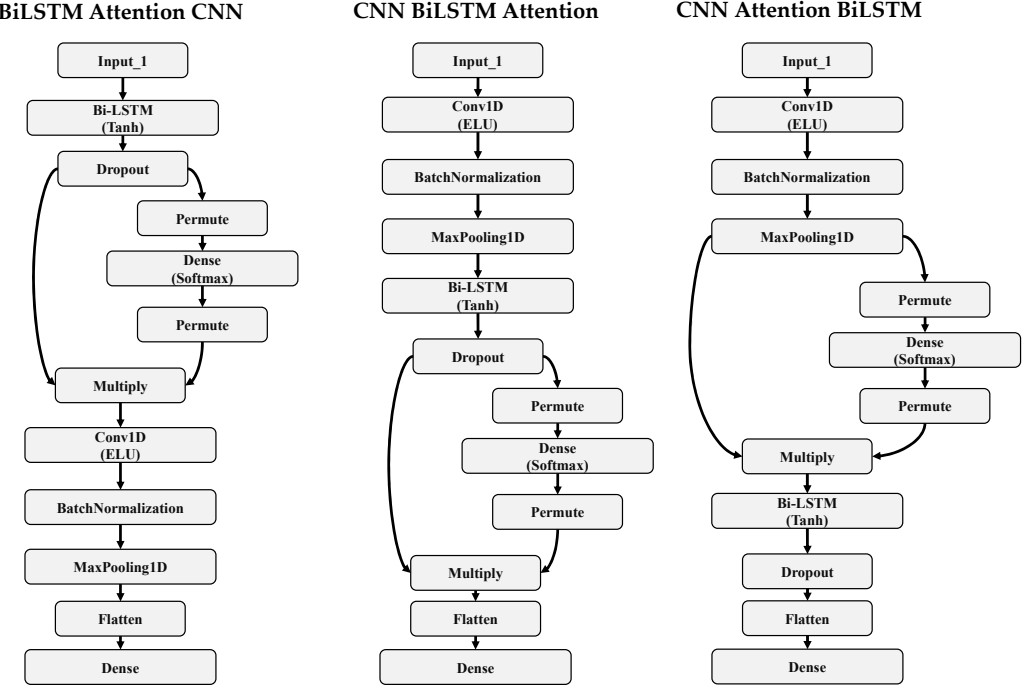

**Figure 1.** Structural differences among the three hybrid models. Source: the authors.

The BiLSTM Attention CNN model initiates with a BiLSTM layer, focusing on capturing the past and future contextual nuances of sequential data. This is succeeded by the Attention mechanism, which gauges the importance across different time steps and concludes with a CNN layer dedicated to local feature extraction and spatial dimension reduction. It is particularly adept for tasks that demand an intricate blend of sequential and spatial insights [19,21]. Conversely, the CNN BiLSTM Attention model commences with a CNN layer, concentrating on the extraction of salient features. These identified features are then channeled through a BiLSTM layer, capturing both preceding and subsequent time-step information. The final layer, the Attention mechanism, emphasizes the most crucial elements within the sequence. This model is optimally suited for tasks, such as sentiment analysis, where the extraction of local features takes precedence over temporal dependencies [19,21]. Lastly, the CNN Attention BiLSTM model starts with the CNN layer for feature distillation, followed closely by the Attention mechanism assessing the significance of various spatial regions. A BiLSTM layer finalizes the model, absorbing sequential data from both past and future contexts. This configuration is ideal for scenarios requiring a keen spatial focus before delving into temporal nuances [19,21].

In essence, while each model offers distinct advantages, the choice invariably depends on the nature of the task and the intrinsic characteristics of the dataset. For instance, the BiLSTM Attention CNN model accentuates sequential patterns, whereas the CNN BiLSTM Attention and CNN Attention BiLSTM models are more inclined toward local feature discernment. Conclusively, hybrid models, integrating the prowess of the CNN, BiLSTM, and the Attention mechanism, emerge as potent tools for a spectrum of NLP and prediction endeavors. Their distinct configurations and nuanced attention implementations mean performance variations across tasks, necessitating a profound understanding for optimal application [19,21].

*2.5. Technical Analysis Methods*

Technical analysis methods are statistical methodologies designed to examine and interpret market data, such as price and volume [22]. These methods identify trends and generate trading signals, serving as crucial tools for traders, investors, and portfolio managers. By utilizing these techniques, these financial experts can significantly enhance their decision-making processes and increase the accuracy of their predictions. Several technical analysis methods exist, each focusing on different aspects of market data. These include trend, momentum, volatility, and volume analysis [23].

## 3. Research Context

*3.1. Research Procedure*

The research process in this study, as outlined in Figure 2, commences with the collection of stock data and proceeds. Subsequently, the performance of different models is compared to determine the most effective model. This study continues with the optimization of model performance through hyperparameter tuning. Lastly, a back-testing rate of return validation is performed, utilizing individual deep learning models.

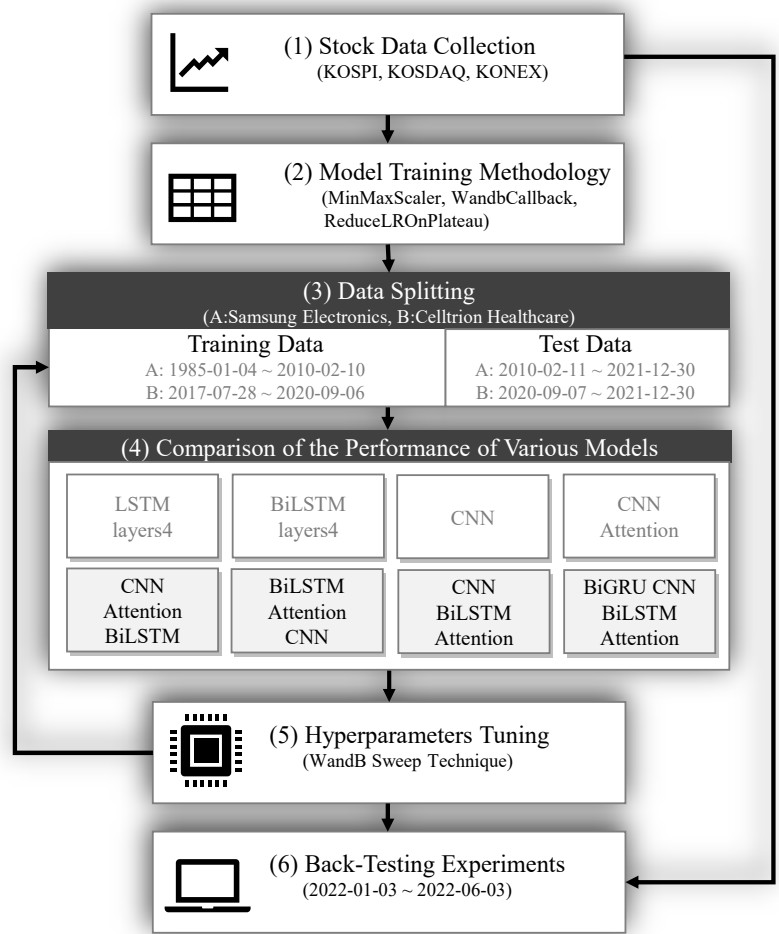

**Figure 2.** Research procedure. Source: the authors.

*3.2. Experimental Environmental*

We set an experimental environment composed of an AMD Ryzen 7 4800H with Radeon Graphics (2.90 GHz processor), 8.00GB RAM, and an NVIDIA GeForce GTX 1650 Ti GPU as personal laptop GPUs. The research was conducted within the VScode programming environment, Keras 2.4.3, TensorFlow 2.3.0, and Python 3.7.13. These tools allowed for multi-NVIDIA GPU support and distributed training.

Stock data were gathered using a 32-bit Anaconda virtual environment. On the other hand, experiments involving deep learning models and research comparing profitability through back-testing methods were executed in a 64-bit Anaconda virtual environment.

Database management was accomplished with MySQL 8.0. JetBrains DataGrip was chosen as the SQL query execution tool over MySQL Workbench due to its greater versatility and user-friendliness.

### 3.3. Stock Data Collection

In this study, we utilized the Kiwoom Securities (hereafter "K") OpenAPI to collect daily stock data from approximately 2546 stocks listed on the KOSPI, KOSDAQ, and KONEX, explicitly focusing on daily data instead of minute-level data. In addition, the K OpenAPI was employed to gather closing prices, and all subsequent experiments were conducted based on these collected data.

The dataset has eight columns: date, code, stock name, closing price, 5-day simple moving average, 10-day simple moving average, 20-day simple moving average, and 40-day simple moving average. In addition, for the comparative analysis of deep learning models and back-testing strategies, a dataset containing five columns—close, open, high, low, and volume—was utilized for implementing buying and selling decisions with deep learning models.

In addition, data regarding managed stocks, non-compliant disclosure stocks, investment caution stocks, investment warning stocks, and investment risk stocks were scraped from the Korea Investor Network Disclosure Channel (KIND) website.

## 4. Methodology

### 4.1. Model Training

We used 9780 data points from Samsung Electronics stock data from 4 January 1985 to 30 December 2021. In addition, 1088 data points from Celltrion Healthcare stock data, covering the period from 28 July 2017 to 30 December 2021, were also employed in the experiments. Samsung Electronics and Celltrion Healthcare are well-established companies with solid reputations in their respective industries. This means their stock data reflects mature market behavior, which can be beneficial when making predictions.

Moreover, the reason for choosing these two items was because at the time of data collection, they were each at the top in terms of market capitalization in the KOSPI and the KOSDAQ, respectively.

For model performance validation experiments, input data were extracted from five column features: closing price (close), trading volume (volume), opening price (open), highest price (high), and lowest price (low). In this study, only the close, volume, open, high, and low columns were extracted as features from the Samsung Electronics and Celltrion Healthcare stock data, and all five columns were normalized to values using MinMaxScaler [24].

Deep learning models were trained using the fit function in the model class API, leveraging the TensorFlow.keras.callbacks library, wandb.keras, and WandbCallback. To monitor the internal state and statistical data of the deep learning models during training, we employed two callback functions: WandbCallback and ReduceLROnPlateau. These callback functions facilitated more effective and precise training of the deep learning models [25].

We employed the WandbCallback function from the wandb.keras library to save the best-performing model and weights based on the lowest validation data loss, as well as to record the performance metrics of the model at each epoch. The validation_steps parameter was set to "5", enabling the full validation dataset to be used every five steps. Due to its significant benefits, we chose the WandbCallback function over the TensorBoard callback function. To optimize model performance, we used the ReduceLROnPlateau callback function, which lowers the learning rate (lr) if there is no improvement or decrease in validation data loss over one generation. Notably, in this study, the default value for the

min_lr variable, which establishes the lower limit for the learning rate, was adjusted from "0" to "0.0001" [26].

We also used the train_test_split function from the Sklearn library to divide the training and testing data at a ratio of 7:3. The experiments were performed with EPOCHS set to 100 and BATCH_SIZE to 32. In this research, we made a concerted effort to implement the AngularGrad optimizer [27] in various deep learning models. Therefore, we meticulously configured the parameters, setting the method angle to "cos," the learning rate to $1 \times 10^{-4}$, beta_1 to 0.9, beta_2 to 0.999, and eps to $1 \times 10^{-7}$. The intent behind this arrangement is to augment the optimization process, thus improving both the performance and generalization capability of the models.

### 4.2. Window Sliding

The window sliding technique is popular in stock price prediction using deep learning models [28]. This method segments a time series of stock prices into smaller, equal-sized windows or segments, which are slid across the data. The primary goal of employing the window-sliding method in deep learning models is to detect key patterns and trends in the data that can be used to generate accurate predictions. Moreover, the window sliding method allows deep learning models to discern temporal dependencies within the data, which is vital in forecasting stock prices [29].

In this study, experiments were carried out using a window size of 1. This strategy divided the time series of stock prices into consecutive, non-overlapping windows of size 1, enabling daily analysis of the stock price data and capturing short-term trends and market fluctuations. Utilizing a window size of 1 allowed us to grasp the daily volatility of the stock prices and detect patterns and trends that might not be observable with other techniques. This approach also facilitated the optimization of predictive models by analyzing the performance of various models.

### 4.3. Deep Learning Models Comparison

We conducted an in-depth evaluation of multiple models to facilitate a comprehensive comparison, encompassing primary and hybrid configurations. The basic models assessed in this research comprised four-layer LSTM networks, CNNs, and four-layer BiLSTM networks. Furthermore, the hybrid models assessed in the study included the CNN Attention hybrid model, the BiLSTM Attention CNN hybrid model, the CNN BiLSTM Attention hybrid model, the CNN Attention BiLSTM hybrid model, and the Bidirectional Gated Recurrent Unit (BiGRU) CNN BiLSTM Attention, hybrid model. We used the same values for the hyperparameters to compare the models.

We utilized the BiGRU CNN BiLSTM Attention model, an innovative deep learning architecture integrating the BiGRU, CNN, and BiLSTM alongside an Attention mechanism. This architecture aims to improve time series data's prediction performance and interpretability by selectively concentrating on relevant parts of the input sequence.

The model starts with an input layer of shape (5, 1), symbolizing a one-time step with five features. Next, a bidirectional GRU layer is implemented to capture both forward and backward dependencies within the input sequence. Next, the BiGRU layer comprises two GRU layers, one processing the input sequence in the forward direction and the other in the reverse direction. Finally, the outputs of the two GRU layers are combined to produce a comprehensive representation of the input sequence.

Next, a 1D convolutional layer with Exponential Linear Unit (ELU) activation is applied to the output of the BiGRU layer. This convolutional layer is designed to identify local patterns and features within the input sequence, thereby extracting relevant temporal information. A batch normalization layer is then used to enhance the model's training stability and convergence speed by normalizing the output of the convolutional layer. Following the normalization, a max pooling layer is employed to reduce the spatial dimensions of the feature maps, thereby enhancing the model's ability to identify and extract robust, reliable features from the data. Following this, a bidirectional LSTM layer is introduced to further

refine the temporal information captured by the model. Like the BiGRU layer, the BiLSTM layer comprises two LSTM layers processing the input sequence in both forward and reverse directions, with their outputs combined. It is applied to the output of the dropout to incorporate the Attention mechanism into the model. The input tensor dimensions are rearranged using the Permute function in this Attention mechanism. A dense layer with a SoftMax activation computes attention weights for each time step, signifying their importance in the sequence.

The second Permute function reshuffles the dimensions of the input tensor (42, 1) based on a specified pattern. The tensor's form is effectively reshaped by rearranging the dimensions, offering flexibility in processing the tensor. The output of the dropout layer, which represents the temporal sequence of input features before the Attention mechanism, then undergoes an element-wise multiplication with these attention weights using the multiply function. The result is a new sequence where each feature is scaled according to its calculated weight or significance. After implementing the Attention mechanism, a Flatten layer is utilized to reshape the output tensor into a 2D tensor. This reshaped tensor is then fed into a dense layer with a linear activation function in the model's final output.

A novel deep learning model, CNN Attention BiLSTM, is proposed in this study, combining a CNN, an Attention mechanism, and BiLSTM for time series prediction. This model seeks to address the complexities associated with deciphering intricate patterns and dependencies in sequential data, especially within the context of financial market applications. The architecture of the proposed model is detailed as follows. (1) Input Layer: The input layer takes in data with a shape of (5, 1), where 5 represents the five input features, including closing, volume, opening, highest, and lowest prices, and 1 signifies the number of time steps. (2) Convolutional Layer: A 1D convolutional layer is implemented to extract local features from the input time series data. The "filters" parameter in the conv1 function represents the dimensionality of the output space. Each filter in a CNN is responsible for a specific type of feature extraction by convolving with the input data. This 1D convolutional layer has filters of 21, a kernel size of 1, and strides of 30. The layer employs the ELU activation function and valid padding. (3) Batch Normalization Layer: Batch normalization is a technique aimed at enhancing the training process and attaining quicker convergence by normalizing the input to each layer. A batch normalization layer is added with an axis of 1 and a momentum of 0.9. (4) Max Pooling Layer: A 1D max pooling layer with a pool size of 1 and strides of 2 is implemented to reduce the spatial dimensions of the input data. This layer aids in decreasing the number of trainable parameters. (5) Attention Mechanism: Proven effective in stock prediction tasks, the Attention mechanism is an integral part of this architecture. It assigns importance to different input time steps. Following the implementation of a 1D max pooling layer, the Attention mechanism comes into play. This mechanism reshapes the input tensor dimensions using the Permute function and calculates attention weights for each timestep. These weights reflect the importance of each timestep within the sequence. This is executed using a dense layer that employs a SoftMax activation function. Then, another Permute function rearranges the input tensor according to a pre-set pattern, allowing for flexible tensor processing. Lastly, the output from the 1D max pooling layer undergoes an element-wise multiplication with these attention weights, yielding a new sequence where each feature's magnitude is adjusted according to its computed significance. (6) Bidirectional LSTM Layer: A BiLSTM layer processes sequential data in both forward and backward directions, enabling the model to capture patterns from past and future data. This layer has 21 units and uses a kernel initializer set to its default value. (7) Dropout Layer: To prevent overfitting and enhance the model's generalization capabilities, a dropout layer with a rate of 0.3 is introduced. This layer randomly omits neurons during the training phase. (8) Flatten Layer: The output tensor from the preceding layer is reshaped into a two-dimensional tensor using a Flatten layer. (9) Output Layer: A dense output layer with a single neuron and a linear activation function is employed for regression tasks. It predicts the target value based on the learned features.

In conclusion, the proposed CNN Attention BiLSTM model integrates the strengths of a CNN, Attention mechanism, and BiLSTM network to predict time series data effectively. As a result, this model is anticipated to offer valuable insights and enhance prediction performance across various financial market applications. Additionally, this streamlined architecture provides an efficient solution for stock price prediction within automated stock trading systems.

*4.4. Hyperparameter Tuning*

This study employed the Sweep methodology from Weights and Biases (WandB), providing a robust and efficient method for hyperparameter optimization specifically designed for deep learning models [30].

The process involves exploring various hyperparameter values to find the most effective combination, which can be time-consuming. To expedite this process, we utilized Bayesian optimization, a method known for its speed in identifying optimal solutions. Moreover, we conducted hyperparameter tuning experiments on four distinct hybrid models. These models included CNN Attention BiLSTM, BiLSTM Attention CNN, CNN BiLSTM Attention, and BiGRU CNN BiLSTM Attention. Each model was analyzed and optimized to ensure the most effective and efficient performance.

Within the BiLSTM layers of these models, the initial default values were replaced with alternative settings:

- recurrent_initializer = tf.random_normal_initializer(mean = 0.2, stddev = 0.05);
- bias_initializer = tf.keras.initializers.HeUniform();
- kernel_initializer = tf.keras.initializers.GlorotNormal().

Employing these parameters in place of the default settings in the BiLSTM layer could enhance model performance and learning efficiency. In deep learning models, selecting suitable weigh properties can significantly influence the attainment of optimum performance. Commonly, the following weight initializations are employed in designing neural networks. (1) Recurrent Initializer: The recurrent_initializer sets up the initial weights of the recurrent layers within a neural network. Using a random normal initializer with specified mean and standard deviation demands a comprehension of these parameters' implications on the initial weights' distribution. The mean = 0.2 parameter determines the center of the normal distribution from which the initial weights are derived. Typically, the mean is set near 0, but in this scenario, it has been set to 0.2, slightly shifting the range of initial weights. By doing so, the model can examine a broader set of initial weight values, potentially facilitating quicker convergence and preventing the model from falling into local minima during training. The stddev = 0.05 parameter defines the breadth or dispersion of the normal distribution employed for generating the initial weights. A lower standard deviation value leads to a tighter distribution of initial weights around the mean. The initial weight values will be nearer to the mean of 0.2. A minor standard deviation, such as 0.05, allows the model to explore a regulated range of initial weight values, helping to avoid issues like vanishing or exploding gradients during training.

In summary, using a random normal initializer with mean = 0.2 and stddev = 0.05 means the initial weights are drawn from a normal distribution centered around 0.2 with a spread of 0.05. This setting could result in more diverse initial weight values, possibly enhancing model convergence and alleviating common training challenges [31].

This study conducted hyperparameter tuning experiments on three models: CNN Attention BiLSTM, BiLSTM Attention CNN, and CNN BiLSTM Attention. A Sweep Configuration approach was used, encompassing five key parameters. (1) BiLSTM_units was tested with three unique values: 11, 16, and 21. (2) Conv1D_activation investigated three activation functions: ReLU, SeLU, and ELU. (3) Conv1D_filters were evaluated with two distinct values: 21 and 31. (4) Conv1D_strides were examined with two stride values: 30 and 40. (5) Dropout was scrutinized with three different settings: 0.2, 0.3, and 0.4.

In addition, the BiGRU CNN BiLSTM Attention model underwent Sweep Configuration with six parameters. The parameters for BiLSTM_units, Conv1D_activation,

Conv1D_filters, Conv1D_strides, and dropout were set identically to those in the Sweep Configuration of the CNN Attention BiLSTM, BiLSTM Attention CNN, and CNN BiLSTM Attention models. However, the distinguishing feature of this model was the introduction of an additional parameter, BiGRU_units, into the Sweep Configuration. Hyperparameter optimization experiments were conducted with the BiGRU_units parameter assigned the values 11, 16, and 21.

*4.5. Back-Testing with Technical Analysis Methods*

We designed a back-testing system that primarily targets acquiring stocks at the opening price at 9 AM on the specified "d" date. This method emphasizes the daily trading of stocks, initiating transactions at the start of the trading day.

Technical analysis methods were applied by firstly excluding stocks classified as administrative issues, unfaithful corporations, investment caution stocks, investment warning stocks, and investment risk stocks from all KOSPI, KOSDAQ, and KONEX stocks. Then, stocks were selected based on a complex formula involving the 5-day, 10-day, 20-day, and 40-day average closing prices and the Bollinger Bands method [32]. Stocks were further filtered based on the "d-1" date closing price's position relative to the lower Bollinger Band. In the last phase of extraction, stocks that satisfied the condition of having an average absolute momentum percentage more significantly throughout 1 to 12 months prior were chosen.

The buying and selling techniques involved using hybrid models with the five-column data (closing price, trading volume, opening price, highest price, and lowest price) from each stock's listing date up to the "d-1" date.

The buying technique involved using four individual hybrid deep learning models with hyperparameters determined through WandB Sweep. These hyperparameters were determined based on the most common values from the top three results with the lowest val_mae values, obtained through WandB Sweep experiments using data from Celltrion Healthcare and Samsung Electronics stocks.

First, the models were used to purchase stocks with the predicted closing price rise by exceeding "1" percent on the "d" date compared to the predicted closing price on the "d-1" date. Then, starting with a capital investment of 10 million KRW, a daily allocation of 2 million KRW was distributed across a range of stocks to promote diversification.

The selling technique involved using the same four individual hybrid deep learning models. Stocks were sold when the predicted closing price for the "d" date was expected to decline by less than "-1" percent compared to the predicted closing price on the "d-1" date. The same deep learning models were used for buying and selling techniques to establish a cohesive trading strategy. Moreover, the hyperparameter values of hybrid models used for selling techniques were the same as those used in the buying techniques.

## 5. Results

*5.1. Deep Learning Models Comparison*

We utilized stock data from Samsung Electronics and Celltrion Healthcare to conduct a thorough performance comparison of eight distinct deep learning models. Each model was trained over 100 epochs, and the primary metrics used for evaluation were validation MAE (Val_MAE) and validation RMSE (Val_RMSE) on the validation dataset. The complete results of this comparative analysis are presented in Table 1, where the outcomes are systematically arranged in ascending order based on their Val_MAE values. Upon careful examination of these results, it becomes evident that the CNN Attention BiLSTM model outperforms the other models, exhibiting the lowest Val_MAE value of 0.00292 for Samsung Electronics and 0.01726 for Celltrion Healthcare. The second most effective model in terms of Val_MAE value is the BiGRU CNN BiLSTM Attention model, with a Val_MAE value of 0.00312 for Samsung Electronics and 0.01733 for Celltrion Healthcare.

**Table 1.** Performance comparison of deep learning models.

| Samsung Electronics | | | | |
|---|---|---|---|---|
| **Model** | **MAE** | **RMSE** | **Val_MAE** | **Val_RMSE** |
| CNN Attention BiLSTM | 0.01253 | 0.02630 | 0.00292 | 0.00520 |
| BiGRU CNN BiLSTM Attention | 0.01241 | 0.02713 | 0.00312 | 0.00586 |
| BiLSTM Attention CNN | 0.01269 | 0.02615 | 0.00313 | 0.00508 |
| Bi LSTM layers 4 | 0.01483 | 0.02929 | 0.00369 | 0.01756 |
| CNN BiLSTM Attention | 0.01297 | 0.02820 | 0.00424 | 0.00630 |
| CNN Attention | 0.01954 | 0.03987 | 0.00557 | 0.01909 |
| CNN | 0.02336 | 0.04687 | 0.01142 | 0.01506 |
| LSTM layers 4 | 0.02594 | 0.04627 | 0.01472 | 0.02630 |
| Celltrion Healthcare | | | | |
| Model | MAE | RMSE | Val_MAE | Val_RMSE |
| CNN Attention BiLSTM | 0.04244 | 0.06151 | 0.01726 | 0.02695 |
| BiGRU CNN BiLSTM Attention | 0.03764 | 0.05414 | 0.01733 | 0.02362 |
| CNN BiLSTM Attention | 0.04524 | 0.06358 | 0.01788 | 0.02972 |
| Bi LSTM layers 4 | 0.03913 | 0.05646 | 0.02110 | 0.03493 |
| BiLSTM Attention CNN | 0.02944 | 0.04253 | 0.02204 | 0.03319 |
| LSTM layers4 | 0.05443 | 0.08092 | 0.02640 | 0.04006 |
| CNN | 0.04866 | 0.06590 | 0.02808 | 0.03818 |
| CNN Attention | 0.18247 | 0.27261 | 0.02991 | 0.04754 |

Source: the authors.

The BiGRU CNN BiLSTM Attention model combines the strengths of BiGRU, CNNs, and BiLSTM layers [33]. While this model is designed to capture spatial patterns and temporal relationships, the complexity of combining multiple layers may introduce additional challenges in model optimization and training. On the other hand, with its streamlined architecture, the CNN Attention BiLSTM model offers a more balanced approach that effectively leverages the strengths of its constituent components while minimizing potential drawbacks associated with increased complexity. Furthermore, this architecture ensures the efficient capture of spatial patterns and temporal relationships and the optimal utilization of the Attention mechanism. Consequently, it leads to superior performance in stock price prediction for automated stock trading systems.

*5.2. Results of Back-Testing with Technical Analysis Methods*

This section presents the results of a comparative study on the rate of return for four individual hybrid models, using back-testing methodologies for evaluation. The back-testing experiments focused on stocks selected exclusively based on technical analysis methods. Four individual hybrid models were used for rate of return validation and comparison: BiGRU CNN BiLSTM Attention, BiLSTM Attention CNN, CNN BiLSTM Attention, and CNN Attention BiLSTM. The configuration of hyperparameters in the models in this study is presented in Table 2. The rate of return was assessed and compared by sequentially testing each of the four models in the back-testing process.

**Table 2.** The configuration of hyperparameters in the models.

| Hyperparameters | | BiLSTM Attention CNN | CNN BiLSTM Attention | CNN Attention BiLSTM | BiGRU CNN BiLSTM Attention |
|---|---|---|---|---|---|
| Conv1D | Filters | 21 | 31 | 31 | 31 |
| | Activation | elu | elu | selu | elu |
| | Kernel Size | 1 | 1 | 1 | 1 |
| | Strides | 40 | 30 | 30 | 40 |
| Batch Normalization | Axis | 1 | 1 | 1 | 1 |
| | Momentum | 0.9 | 0.9 | 0.9 | 0.9 |
| MaxPooling1D | Pool Size | 1 | 1 | 1 | 1 |
| | Strides | 2 | 2 | 2 | 2 |
| Bidirectional LSTM | Units | 21 | 21 | 21 | 21 |
| Bidirectional GRU | Units | None | None | None | 16 |
| Dropout | Ratio | 0.2 | 0.3 | 0.2 | 0.2 |
| Output | Units | 1 | 1 | 1 | 1 |
| Dense layer | Activation | linear | linear | linear | linear |
| Batch Size | | 32 | 32 | 32 | 32 |
| Epochs | | 100 | 100 | 100 | 100 |
| Optimizer | | AngularGrad ("cos") | AngularGrad ("cos") | AngularGrad ("cos") | AngularGrad ("cos") |
| Learning Rate (AngularGrad) | | $1 \times 10^{-4}$ | $1 \times 10^{-4}$ | $1 \times 10^{-4}$ | $1 \times 10^{-4}$ |
| loss function | | MAE | MAE | MAE | MAE |
| Performance metrics | | MAE, RMSE | MAE, RMSE | MAE, RMSE | MAE, RMSE |

Source: the authors.

We employed the same technical analysis methods and models for both buying and selling techniques, enabling an accurate comparison across the four hybrid models based on their performance. The back-testing period extended from 3 January 2022 to 3 June 2022, covering approximately five months. During this period, a comparative study on the rate of return was conducted for the four hybrid models. The initial investment capital was set at KRW 10 million, with around KRW 2 million invested in a single stock, and the remaining funds were allocated in approximately KRW 2 million increments across various stocks to achieve diversification. As detailed in Table 3, we employed technical analysis methods to extract and analyze stocks using a CNN Attention BiLSTM model. The purchasing was restricted to stocks exhibiting a ratio value greater than "1" percent. About KRW 2 million was allocated to each of the 12 selected stocks to encourage diversification. Due to the investment limit set at KRW 2 million per stock, only 31 shares of stock code 041510, SM Corporation, were bought for KRW 1,946,800 on 10 January 2022, at 9 AM. This model yielded an approximate 5% increase in the rate of return over five months of back-testing, and the final balance on 3 June 2022 stood at KRW 10,511,708. For the BiGRU CNN BiLSTM Attention model, the back-testing results over five months showed an approximate 1% increase in the rate of return, and the final balance on 3 June 2022, stood at KRW 10,095,888. For the BiLSTM Attention CNN model, back-testing revealed an approximate 2% increase in the rate of return. As a result, the final balance on 3 June 2022, came to KRW 10,217,791. Lastly, the CNN BiLSTM Attention model led to an approximate 4.5% rise in the rate of return over five months. Consequently, the final balance on 3 June 2022, amounted to KRW 10,456,241.

**Table 3.** Results of back-testing.

| Code | Code Name | Rate | Purchase Price (KRW) | Holding Amount | Buy Date (YYYYMMDD) | Sell Date (YYYYMMDD) |
|---|---|---|---|---|---|---|
| **CNN Attention BiLSTM** | | | | | | |
| 041510 | SM | 7.34 | 62,800 | 31 | 20220110 | 20220117 |
| 037270 | YG PLUS | −3 | 7330 | 272 | 20220110 | 20220111 |
| 037270 | YG PLUS | −3.73 | 6930 | 288 | 20220113 | 20220117 |
| 002380 | KCC | 1.81 | 310,500 | 6 | 20220217 | 20220223 |
| 318010 | Pharmsville | −4.4 | 13,300 | 150 | 20220228 | 20220302 |
| 210120 | Victory Contents | 34.41 | 13,800 | 144 | 20220316 | 20220324 |
| 088290 | Ewon Comfortech | 1.64 | 8320 | 240 | 20220415 | 20220420 |
| 336060 | Wavus | −3.65 | 2660 | 751 | 20220419 | 20220420 |
| 085370 | Lutronic | −2.51 | 22,350 | 89 | 20220511 | 20220512 |
| 067830 | Savezone I&C | 0.32 | 3325 | 601 | 20220511 | 20220525 |
| 002420 | Century Corporation | −0.28 | 9910 | 201 | 20220516 | 20220519 |
| 001080 | Manho Rope & Wire | −1.87 | 25,100 | 79 | 20220516 | 20220519 |
| Profit: KRW 511,708 (approximately 5% rate of return) | | | | | | |
| **BiGRU CNN BiLSTM Attention** | | | | | | |
| 041510 | SM | 7.34 | 62,800 | 31 | 20220110 | 20220117 |
| 037270 | YG PLUS | −3 | 7330 | 272 | 20220110 | 20220111 |
| 037270 | YG PLUS | −3.73 | 6930 | 288 | 20220113 | 20220117 |
| 247540 | EcoPro BM | 2 | 320,000 | 6 | 20220128 | 20220207 |
| 002380 | KCC | 1.81 | 310,500 | 6 | 20220217 | 20220223 |
| 318010 | pharmsville | −4.4 | 13,300 | 150 | 20220228 | 20220302 |
| 088290 | Ewon Comfortech | 1.64 | 8320 | 240 | 20220415 | 20220420 |
| 085370 | Lutronic | −2.51 | 22,350 | 89 | 20220511 | 20220512 |
| 067830 | Savezone I&C | 0.32 | 3325 | 601 | 20220511 | 20220525 |
| 000060 | Meritz Fire | 6.28 | 37,250 | 53 | 20220513 | 20220520 |
| 002420 | Century Corporation | 1.36 | 9750 | 205 | 20220513 | 20220516 |
| 001080 | Manho Rope & Wire | −1.87 | 25,100 | 79 | 20220516 | 20220519 |
| Profit: KRW 95,888 (approximately 1% rate of return) | | | | | | |
| **BiLSTM Attention CNN** | | | | | | |
| 200580 | Medyssey | −1.25 | 10,300 | 194 | 20220113 | 20220114 |
| 028050 | Samsung Engineering | 10.46 | 20,900 | 95 | 20220128 | 20220208 |
| 036620 | Gamsung Corporation | 4.71 | 2000 | 1000 | 20220204 | 20220208 |
| 036620 | Gamsung Corporation | −1.58 | 1920 | 1041 | 20220415 | 20220418 |
| 004890 | DONGIL INDUSTRIES | −2.93 | 169,500 | 11 | 20220512 | 20220513 |
| 002420 | Century Corporation | 1.36 | 9750 | 205 | 20220513 | 20220516 |
| Profit: KRW 217,791 (approximately 2% rate of return) | | | | | | |
| **CNN BiLSTM Attention** | | | | | | |
| 041510 | SM | 7.34 | 62,800 | 31 | 20220110 | 20220117 |
| 037270 | YG PLUS | −3 | 7330 | 272 | 20220110 | 20220111 |
| 037270 | YG PLUS | −3.73 | 6930 | 288 | 20220113 | 20220117 |
| 092200 | Dic | −2.77 | 5000 | 400 | 20220216 | 20220218 |

**Table 3.** *Cont.*

| | CNN Attention BiLSTM | | | | | |
|---|---|---|---|---|---|---|
| Code | Code Name | Rate | Purchase Price (KRW) | Holding Amount | Buy Date (YYYYMMDD) | Sell Date (YYYYMMDD) |
| 002380 | KCC | 1.81 | 310,500 | 6 | 20220217 | 20220223 |
| 318010 | Pharmsville | −4.4 | 13,300 | 150 | 20220228 | 20220302 |
| 210120 | Victory Contents | 34.41 | 13,800 | 144 | 20220316 | 20220324 |
| 088290 | Ewon Comfortech | 1.64 | 8320 | 240 | 20220415 | 20220420 |
| 336060 | Wavus | −3.65 | 2660 | 751 | 20220419 | 20220420 |
| 085370 | Lutronic | −2.51 | 22,350 | 89 | 20220511 | 20220512 |
| 067830 | Savezone I&C | 0.32 | 3325 | 601 | 20220511 | 20220525 |
| 002420 | Century Corporation | −0.28 | 9910 | 201 | 20220516 | 20220519 |
| 001080 | Manho Rope & Wire | −1.87 | 25,100 | 79 | 20220516 | 20220519 |
| Profit: KRW 456,241 (approximately 4.5% rate of return) | | | | | | |

Source: the authors.

In all cases, the final profit amount (sum_valuation_profit) incorporated the calculation of fees and taxes deducted from the sum of the initial investment capital and the profit amounts for each stock. As outlined in Figure 3, the study spanned from 3 January 2022 to 3 June 2022, focusing on South Korea's two leading stock market indices, the KOSPI and the KOSDAQ. These indices, representing diverse companies and market segments, experienced a bearish market for approximately five months. Despite the market downturn, the study conducted back-testing using four hybrid models—CNN Attention BiLSTM, BiGRU CNN BiLSTM Attention, BiLSTM Attention CNN, and CNN BiLSTM Attention—to verify the rate of return on investment. Notably, all four models displayed increased rates of return, even amid the bearish market conditions.

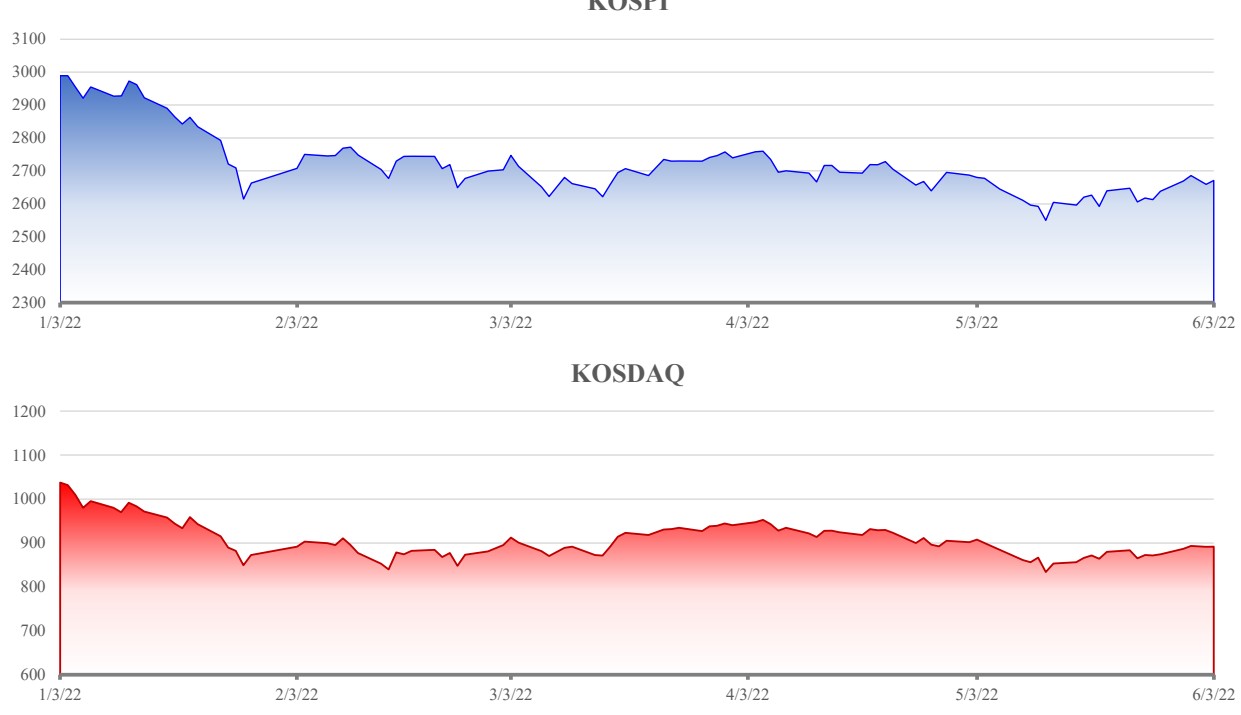

**Figure 3.** Historical performance of the KOSPI and KOSDAQ indices. Source: NAVER Financial.

A streamlined technical analysis approach was crucial to this achievement, allowing for a steady return rate despite the bear market. The methodology, combining rigorous analysis and innovative techniques, effectively offset the potential negative impacts of market downturns, demonstrating its robustness and adaptability in generating consistent profits.

## 6. Discussion

### 6.1. Discussion of Findings

This study examined Samsung Electronics and Celltrion Healthcare stock data using eight deep learning models, among which the CNN Attention BiLSTM outperformed the others. Furthermore, by implementing a back-testing method to verify returns, the CNN Attention BiLSTM model displayed the highest prediction accuracy and returns among the four hybrid models considered. This finding underscores the efficacy of the CNN Attention BiLSTM model in accurately predicting stock market trends and consistently generating profits, emphasizing the value of leveraging advanced deep learning techniques. In this study, utilizing the back-testing method over five months resulted in the following projected timelines. When analyzing the collected data for approximately 2546 stocks from KOSPI, KOSDAQ, and KONEX solely using deep learning techniques, the anticipated time frame was roughly 800 days. In contrast, integrating technical analysis with deep learning analysis significantly reduced the time required, with experimental results indicating a timeline of approximately 2–3 days. Hence, solely analyzing all stocks through deep learning methods proves to be impractical. It becomes necessary to incorporate technical analysis to facilitate feasible back-testing methods, enabling the purchase of potentially rising stocks on the following day.

Theoretically, this study contributes to the existing literature using domestic stock data concerning automated trading systems, an area with relatively few studies [5]. Moreover, it introduced a lightweight technical analysis method that allows conducting analyses using deep learning models and executing buy and sell orders, even in an average GPU environment. This approach effectively addresses the temporal limitations of algorithms that rely solely on deep learning models.

From a practical standpoint, this research offers a cost-efficient and highly accessible approach to developing automated stock trading systems that use machine learning and deep learning. For example, some larger investment companies or institutions use high-priced, ultra-fast GPUs to handle the rapid fluctuations of the stock market by performing complex calculations in a very short time. However, while these systems offer fast response times and high processing capabilities, they also entail considerable costs, increasing the initial investment cost.

In contrast, the system proposed in this study can function effectively on an average laptop, significantly reducing the initial investment cost and widening the range of potential participants in its development and operation. Its flexible design allows the application of various technical analysis techniques, catering to individual investor preferences and objectives. In addition, the back-testing feature enables investors to make informed, rational investment decisions that align with their unique investment styles and risk tolerances.

### 6.2. Limitations and Future Research

In this study, we have identified several limitations. First, our approach primarily relied on lightweight analysis methods for stock price prediction. Incorporating a wider spectrum of data sources, from stock prices to sentiment analysis, could further enhance accuracy and potential returns on investment. Second, the quantitative strategies we proposed represent only one dimension of available investment methods. Their efficacy might be influenced by individual investor characteristics, such as risk tolerance and time horizons. Thus, a diversified portfolio that merges quantitative with other investment strategies is advisable. Third, our study's exclusive focus on the Korean stock market might limit its generalizability to other major markets, such as Japan and Hong Kong. Broader

comparative research across different markets is essential to validate our findings' applicability. Fourth, we explored specific configurations of the machine learning frameworks: LSTM, BiLSTM, and CNN. Describing these as distinct models might be misleading, and other prominent machine learning frameworks were not included in our focus. Future research might benefit from a broader exploration of other machine learning architectures. In conclusion, while this study offers an efficient approach to automated stock trading, a more holistic strategy encompassing diverse techniques, extensive data sources, and risk management is recommended. Future studies should consider these factors to stay adaptive and effective in the dynamic world of stock trading.

**Author Contributions:** Methodology, S.Y.; supervision, S.-B.Y.; writing—original draft preparation, S.Y.; writing—review and editing, S.-H.Y. All authors have read and agreed to the published version of the manuscript.

**Funding:** This work was supported by the government of the Republic of Korea (MSIT) and the National Research Foundation of Korea (NRF-2022K2A9A2A11097154, FY2022).

**Data Availability Statement:** The outcomes of the comparative analysis, which encompassed eight deep learning models utilized in this research, along with the findings from the WandB Sweep experiment, are presented on the subsequent page: https://wandb.ai/seongjae-yoo/projects, accessed on 12 September 2023.

**Conflicts of Interest:** The authors declare no conflict of interest.

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
