# Peer review of "The Design of an Intelligent Lightweight Stock Trading System Using Deep Learning Models: Employing Technical Analysis Methods"

_systems, doi:10.3390/systems11090470_

Round 1

Reviewer 1 Report

The authors present interesting work on stock price prediction. They use recurrent neural networks both individually and in hybrid form.

A few questions and suggestions are then given:

- From a formal point of view, the document is well structured and well written. The only point to note is that it could be useful to indicate the source (e.g., own elaboration) in the tables and figures.

- In the introduction section, the authors comment on the need to incorporate automated stock trading systems. To reinforce this idea, Arribas et al. (2019) can be quoted as saying that these systems can help to incorporate investors' preferences on ESG objectives for different stocks.

  Arribas, I., Espinós-Vañó, M.D., García, F., Oliver, J. 2019. Defining socially responsible companies according to retail investors’ preferences. Entrepreneurship and Sustainability Issues, 7(2), 1641-1653. http://doi.org/10.9770/jesi.2019.7.2(59)

 - In section 2-3 a brief explanation of the different models proposed, as well as the hybrid model, would be useful. Although these are known models, at least some kind of graphical scheme for the hybrid model would be useful. This may help to clarify the structure of the model.

- In section 5, why has MAPE not been used as a measure of prediction error?

Reviewer 2 Report

This study investigates automated trading systems using existing ML models on the Korean stock market. My major concerns are listed below.

1. The introduction of ML models is insufficient with too many assumed knowledge. In fact, it is simply surprising to note that the authors fully spell GPU in their abstract, but assumes that the readers understand CNN, BiLSTM and other abbreviations of ML models. There should be a standalone (sub-)section to at least provide an overview of the chosen ML models with justifications.

2. The dataset is limited to the Korean market. It may be more interesting to compare the performance on other highly relevant markets, such as the Japanese and/or Hong Kong stock markets. At the very least, the results can provide robustness check to the empirical findings.

3. Some descriptions are not accurate. For instance, in the abstract, it states that "we analyzed the performance of eight deep learning models: LSTM (4 layers), CNN, BiLSTM (4 layers), CNN Attention, BiGRU CNN BiLSTM Attention, BiLSTM Attention CNN, CNN BiLSTM Attention, and CNN Attention BiLSTM." Many of those claimed models are very similar to each other, which are essentially based on three major ML frameworks. LSTM, BiLSTM and CNN. Instead of claiming 8 models, the authors may consider more accurate descriptions, such as "specifications". Also, there is insufficient justification (as mentioned above) why other popular ML frameworks are not considered in this paper.

4. There are zero figures in this paper. At the very least, the authors can plot the Korean stock indices over time, to visualize their datasets. They should also visualize with an example to demonstrate how a buy/sell trade is completed with real data.

Many of the sentences can read better, and descriptions may be more concise.

Round 2

Reviewer 2 Report

Thank you for revising the paper according to my comments. There are no further issues identified.